# Diabetes Affects the Relationship between Heart Rate Variability and Arterial Stiffness in a Gender-Specific Manner

**DOI:** 10.3390/jcm11174937

**Published:** 2022-08-23

**Authors:** Carla Serra, Alessandro Sestu, Veronica Murru, Giulia Greco, Matteo Vacca, Angelo Scuteri

**Affiliations:** 1Internal Medicine Unit, University Hospital Monserrato, Azienda Ospedaliero-Universitaria di Cagliari, 09123 Cagliari, Italy; 2Post Graduate Medical School of Internal Medicine, University of Cagliari, 09124 Cagliari, Italy; 3Department of Medical Sciences and Public Health, University of Cagliari, 09124 Cagliari, Italy

**Keywords:** arterial stiffness, pulse wave velocity, heart rate variability, gender, diabetes

## Abstract

**Background and Aims:** Heart rate variability (HRV), i.e., the beat-by-beat fluctuations in heart rate (HR) reflecting the autonomic nervous system balance, is altered in patients with diabetes. This has been associated with arterial aging (stiffer arteries) and differs in men and women. The present study hypothesized that the impact of HRV on arterial aging, indexed as carotid–femoral pulse wave velocity (PWV), differs in a gender-specific manner and is affected by diabetes mellitus. **Method:** A total of 422 outpatients (187 women and 235 men) were studied. PWV was measured using the validated SphygmoCor device (AtCor Medical). Time-domain and frequency-domain parameters were measured to assess HRV. **Results:** The prevalence of diabetes was 30.8% with a slight, but nonsignificant, greater prevalence in men. Both age and SBP were independent determinants of PWV in each of the four groups (men and women with or without diabetes). Low-frequency activity was inversely correlated with PWV. It was greater in women without diabetes, but it was not significant in men regardless of the presence of diabetes. **Conclusions:** Beyond age, blood pressure, and diabetes, impaired cardiac autonomic function assessed by determination of HRV was significantly associated with arterial aging. The association between lower sympathetic and parasympathetic activity and stiffer arteries was significant in women, but not in men.

## 1. Introduction

An aging population is accompanied by emerging cognitive impairment [1] and a dramatic increase in the number of years lived with disability [2]. Searching for markers of aging [3], arterial stiffness, indexed as carotid–femoral pulse wave velocity (PWV), is considered a good proxy of large artery aging, from the early (accelerated) vascular aging (EVA) [4,5] to the “lower than average” arterial aging, i.e., healthy vascular aging (HVA) [5]. A greater PWV has been associated with greater blood pressure variability [6], multiple organ damage [7], increased cardiovascular (CV) morbidity and mortality [8], and cognitive impairment [9].

The prevalence of type 2 diabetes mellitus is increasing worldwide and significantly impacts on CV mortality [10]. Previous studies have shown that arterial stiffness is increased in people with diabetes [11].

To better characterize factors contributing to the modulation of arterial stiffness, we studied heart rate variability (HRV), i.e., the beat-by-beat fluctuations in heart rate (HR). HRV reflects the autonomic nervous system balance on the heart and is an indicator of CV risk [12] and mortality [13]. In general, increased cardiac sympathetic drive is detrimental, while increased cardiac parasympathetic activity is cardioprotective [14].

Alterations in HRV are common in diabetes [15] and have been linked with the severity of diabetes [15]. Of note, in diabetic patients, cardiac autonomic dysfunction has been associated with stiffer arteries independent of age and BP levels [16].

The aim of the present study was to characterize gender-differences with regard to the impact of HRV on arterial stiffness and whether they differ according to the presence of diabetes.

## 2. Methods

### 2.1. Study Population

The study population consisted of subjects who visited our Outpatient Clinic for evaluation of their blood pressure levels. Therefore, the study population resulted in a combination of subjects who resulted hypertensive and/or were undergoing a periodic follow-up of their condition and of subjects who were “unaffected”/healthy, often undergoing the visit as requested by their fitness center. Patients were excluded if they had cancer, acute myocardial infarction in the previous 6 months, hepatic or cardiac failure, serum creatinine ≥2 mg/dL, secondary hypertension, or thyroid disease. Additional exclusion criteria were a previous large vessel stroke and atrial fibrillation, since its presence interferes with the accuracy of aorta stiffness measurement.

After informed consent was given, all patients underwent a medical history, a clinic visit, urinalysis, electrocardiography, and traditional CV risk factor measurements.

We analyzed 422 patients presenting complete measurements of all the parameters investigated. All the measurements were collected at the same visit. These patients will represent the study population of the current study.

Hypertension was defined as the intake of antihypertensive medications and/or if the average of three blood pressure (BP) measurements taken at 5 min interval was ≥140 for SBP or ≥90 mm Hg for DBP. Diabetes was defined as the previous diagnosis and/or use of antidiabetic medications. Body mass index (BMI) was calculated as body weight (kg)/height (m)^2^.

### 2.2. Assessment of Pulse Wave Velocity (PWV)

Aorta stiffness was assessed non-invasively by the carotid–femoral PWV. PWV was measured using the validated SphygmoCor device (AtCor Medical, Australia), whose validation and reproducibility have been previously published [17]. Pulse transit time was determined as the average of 10 consecutive beats. Transit time between carotid and femoral pressure waves was calculated using the foot-to-foot method. The distance traveled by the pulse wave was measured over the body surface subtracting the carotid location–sternal notch distance from the sternal notch–femoral site distance.

### 2.3. HRV Parameters

HRV was measured with the AtCor Medical HRV software to assess sympathetic/parasympathetic autonomic function. A standard 2 min electrocardiogram recording was performed with the patient in the supine position, with a regular and calm breathing pattern (5 min resting study) in a quiet room.

Time domain measures are the mean and standard deviations of RR intervals recorded by the continuous electrocardiogram, where NN intervals represent all the RR intervals. The time domain of HRV was evaluated using the root mean square of successive differences between normal heartbeats (RMSSD) (m/s), which provides an estimate of the short-term components of HRV; the lower this vagal index, the greater the CV burden and the percentage of adjacent NN intervals that differ from each other by more than 50 m (pNN50) (%), which reflects alterations in autonomic function that are primarily vagally mediated and that are virtually independent of circadian rhythms.

The frequency domain of HRV consists of the spectral analysis of a series of consecutive RR intervals in order to quantify sympathetic and vagal influences on the heart. The evaluated parameters were:−power spectral density at the high-frequency (HF) range (0.15–0.4 Hz);−power spectral density at the low-frequency (LF) range (0.04–0.15 Hz);−the LF/HF ratio.

Vagal activity is the major contributor to the HF component and to the time-domain parameters, while LF reflects both sympathetic and vagal activity; the ratio LF/HF is considered to mirror sympathovagal balance [13].

### 2.4. Statistical Analysis

All analyses were performed using SAS University Edition.

PROC UNIVARIATE was adopted to test for normality. Both the Kolmogorov–Smirnov test and Shapiro–Wilk test revealed that the measured variables were not normally distributed; therefore, differences between men and women were compared by the Wilcoxon rank sum test or Mann–Whitney U test (PROC NPAIR1WAY wilcoxon).

Multiple linear regression models were constructed to identify the association of HRV with PWV—separately in men and women, with and without diabetes—independent of age, hypertension, and BMI. SBP, DBP, HR, RMSDD, LF, HF, and LF/HF were all introduced as covariates. Backward elimination was adopted to achieve a more parsimonious model.

An ANCOVA analysis was employed to test for interaction between sex and diabetes.

A two-sided *p* value < 0.05 indicated statistical significance.

## 3. Results

The study population consisted of 422 patients (187 women and 235 men). The prevalence of diabetes was 30.8% with a slight, but nonsignificant, greater prevalence in men.

As illustrated in Table 1, men were older, more frequently hypertensive, with greater BP and stiffer arteries than women.

Multiple linear regression models were constructed to identify the significant determinants of arterial stiffness, indexed as PWV, according to sex and diabetic status. Regression model were adjusted for age, hypertension, and BMI. SBP, DBP, HR, RMSDD, LF, HF, and LF/HF were all introduced as covariates. Backward elimination of nonsignificant variables was adopted to achieve a more parsimonious model.

The regression coefficient (with standard error) between the statistically significant factors in multivariable regression models and PWV are illustrated in Table 2. Both age and SBP were independent determinants of PWV in each of the four groups (men and women with or without diabetes) (Table 2). LF activity was inversely and significantly correlated with PWV in women, but not in men.

To identify whether the impact of LF on PWV differed in men and women with or without diabetes, an ANCOVA analysis was conducted with interaction terms: sex × LF (does the correlation between LF and PWV differ in men and women?); diabetes × LF (does the correlation between LF and PWV differ in diabetic and nondiabetic patients?); sex × diabetes × LF (does the correlation between LF and PWV differ in men and women depending on their diabetic status?) (Figure 1).

## 4. Discussion

In the present study, we showed that beyond age, blood pressure, and diabetes, impaired cardiac autonomic function assessed by determination of HRV was significantly associated with abnormal PWV. The association between lower sympathetic and parasympathetic activity and stiffer arteries was affected by the presence of diabetes, and it was significant in women, but not in men.

High sympathetic activity has been associated with stiffer arteries in patients with and without diabetes [18]. This has been attributed, at least partially, to an increase in heart rate per se. However, in our study the significant association between lower LF activity and stiffer arteries was independent of heart rate levels.

A previous study reported that in diabetic patients, the presence of cardiac autonomic neuropathy was associated with reduced aortic distensibility [19], an association that persisted significantly after controlling for the duration of diabetes [19]. Of note, the study selected diabetic patients without hypertension or macrovascular disease. Additionally, autonomic dysfunction was not further characterized with regards to specific domains of HRV.

Factors modulating and/or impacting on HRV, arterial stiffness, and their relationship are complex. They include smoking, dyslipidemia, level of physical activity, comorbidities, medication for diabetes, and coexisting disease. The TODAY Study reported that glycemic control in younger diabetic subjects was associated with greater sympathetic activity (LF/HF) and stiffer arteries [16]. Additionally, glycemic and blood pressure controls were significantly associated with a worsening of HRV and arterial stiffening over time [20,21].

Gender differences have been reported for several conditions affecting arterial stiffness and/or the autonomic control of the heart: BP [6,22] and heart rate variability [23], walking speed [24], insulin resitance [25], and body weight and shape [26,27].

Concerning the observed gender differences in the association between HRV, namely lower low-frequency activity and PWV, it has been previously reported that, despite greater resting HR in women than in men, heart rate does not have the same predictive power for CV mortality and morbidity in women as it does in men [28,29]. There have been previously reports about gender differences in the “structure” of heart rate variability: women show less power in the low-frequency domain and greater power in the high-frequency domain than men, suggesting that the autonomic control of the heart and heart rate is characterized by a relative dominance of vagal and parasympathetic activity—despite greater mean HR—in women and by a greater sympathetic activity—despite lower resting HR—in men [30].

Characterization of mechanisms underlying the observed gender differences in the association between the low-frequency domain of HRV and arterial stiffness is beyond the design of the present study. A better characterization of the underlying mechanism may improve management and therapeutic indications and pharmacological treatment to reduce the CV burden in diabetes.

The present study has some limitations. The first is the cross-sectional design that does not allow determination of a causal relationship between PWV and cardiac autonomic. Another limitation is that we did not adequately characterize metabolic control in diabetic patients nor the full spectrum of drugs taken by the recruited patients. Given the high percentage of patients with hypertension, several medications (beta- and alfa-blockers, for instance) could interfere with HRV and with its association with arterial stiffness.

To summarize, the results of the present study suggest that the impact of HRV, i.e., the differences in the autonomic regulation of mean HR, on PWV, a parameter that reflects arterial aging, differs in men and women with or without diabetes. Research on HRV needs to emphasize and report sex differences in healthy as well as in population affected by CV conditions.

## Figures and Tables

**Figure 1 jcm-11-04937-f001:**
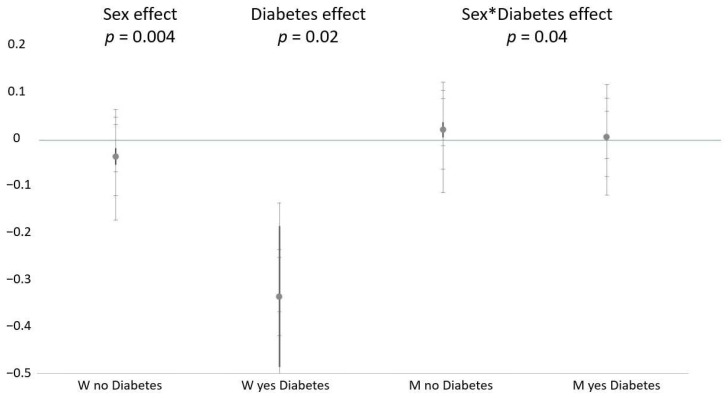
Plots the slopes of the regression between LF activity, and PWV differed according to the presence of diabetes (significant interaction diabetes × LF). Additionally, LF activity showed a significant negative correlation with PWV in women (significant interaction sex × LF). The impact of diabetes on such a correlation differed in men and women (significant interaction sex × diabetes × LF); in fact, it was greater in women without diabetes, but it was not significant in men regardless of the presence of diabetes.

**Table 1 jcm-11-04937-t001:** Characteristic of study population according to sex (median values and interquartile range).

		Women (185)	Men (237)	Gender Difference *p*
Age (years)	61.0 (20.0)	60.0 (27)	62.5 (19)	0.05
Women (%)	43.8	---	--	
Hypertension (%)	62.1	51.9	70.0	0.001
Diabetes mellitus (%)	30.8	26.0	34.6	0.06
BMI (Kg/m^2^)	27.1 (7.1)	26.8 (8.3)	27.5 (6.4)	0.62
SBP (mmHg)	137.0 (23.0)	134.0 (23.0)	139.0 (19.0)	0.01
DBP (mmHg)	77.0 (14.0)	75.0 (16.0)	79.0 (14.0)	0.001
HR (bpm)	67.0 (16.0)	67.0 (16.0)	68.0 (16.0)	0.86
pNN50	2.0 (10.8)	2.2 (12.5)	2.0 (10.0)	0.75
rmSDD	24.6 (27.1)	25.0 (25.9)	24.5 (26.1)	0.72
LF	727.0 (259.0)	788.0 (288.0)	734.0 (234.0)	0.23
HF	273.0 (258.0)	292.0 (288.0)	266.0 (234.0)	0.23
LF/HF ratio	2.7 (2.5)	2.4 (2.6)	2.8 (2.5)	0.23
PWV (m/s)	9.3 (3.1)	8.9 (2.6)	9.8 (3.4)	0.0001
Antihypertensive medication	51.6	42.4	60.1	0.01
Antidiabetic medication	27.8	24.3	30.3	0.05
Lipid-lowering therapy	14.7	11	17.6	0.05

BMI = body mass index; SBP = systolic blood pressure; DBP = diastolic blood pressure; HR = heart rate; pNN50 = percentage of adjacent NN intervals that differ from each other by more than 50 m; rmSDD = root mean square of successive differences between normal heartbeats; LF = low-frequency domain in heart rate variability; HF = high-frequency domain in heart rate variability; and PWV = pulse wave velocity.

**Table 2 jcm-11-04937-t002:** Independent significant determinants of PWV in women and men with or without diabetes, identified by multiple regression analysis (regression beta coefficient +/− standard error) ^+^.

	Women NO Diabetes (*n* = 136)		Women YES Diabetes (*n* = 49)		Men NO Diabetes (*n* = 155)		Men YES Diabetes (*n* = 82)	
Age	3.72 ± 0.97	0.0002	12.22 ± 4.89	0.02	9.32 ± 1.38	0.0001	16.78 ± 3.76	0.0001
SBP	5.72 ± 1.13	0.0001	7.48 ± 2.82	0.02	5.24 ± 1.55	0.001	5.87 ± 3.15	0.05
LF	−0.039 ± 0.018	0.05	−0.337 ± 0.150	0.03	0.018 ± 0.016	0.28	0.002 ± 0.005	0.72
Model R^2^	0.484		0.489		0.479		0.325	

SBP = systolic blood pressure; LF = low-frequency domain in heart rate variability. ^+^ Controlling for age, hypertension, BMI, SBP, DBP, HR, RMSDD, LF, HF, and LF/HF.

## Data Availability

The Dataset Include a Considerable Number of Patients, so According to the Policy at the Time of Ethical Approval, No Public Data Are Available “Automatically”.

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
