# Peer review of "Diabetes Affects the Relationship between Heart Rate Variability and Arterial Stiffness in a Gender-Specific Manner"

_jcm, 2022, doi:10.3390/jcm11174937_

Round 1

Reviewer 1 Report

The author adequately taken into account all the points made in my report.

Author Response

We thank the Reviewer for the appreciation of oru work

Reviewer 2 Report

I have reviewed the new submission of this manuscript and see that the authors are trying to refine the statistical section of their manuscript. However, I still see significant shortcomings in this section. Perhaps the authors need to involve a professional statistician to help refine the statistical processing of the results.

 Namely:

1. The authors noted that "Both Kolmogorov–Smirnov test and Shapiro–Wilk test revelaed that the measured variables were no normally distributed" and at the same time noted that "Data are presented as mean ± SD." It must be remembered that the "average arithmetic is used to represent scale variables with a normal distribution" and that "Standard deviation is intended to describe samples with a normal distribution and is not adapted to non-normal distributions." In a non-parametric distribution, other formats for representing quantitative data must be used: "The median, along with quartiles, is used to represent discrete variables or non-normally distributed quantitative continuous variables".

2. The following statement of the authors also raises questions: "differences in mean values ​​between men and women were compared by non-parametric Wilcoxon test". First, with a nonparametric distribution, one does not have to talk about mean values ​​(see Remark 1). Secondly, the Wilcoxon test can also be used to check the differences between two samples of paired measurements (Wilcoxon signed-rank test), and to assess the differences between two independent samples in terms of the level of any trait measured quantitatively (The Wilcoxon rank-rum test or Mann-Whitney U test). Therefore, it is necessary to indicate which of the Wilcoxon tests the authors used.

3. You should also consider the following feature of non-parametric statistical methods: "The wider applicability and increased robustness of non-parametric tests comes at a cost: in cases where a parametric test would be appropriate, non-parametric tests have less power. In other words, a larger sample size can be required to draw conclusions with the same degree of confidence".

In other words, when evaluating differences between groups, p-values ​​for non-parametric calculations should be markedly larger than when using Student's t-test. If we compare the results of the authors' calculations in the first version of the article and in the submitted manuscript, then such differences are not always noted. For example, the differences for the LF / HF ratio between men and women for were p=0.40 according to the Student's test and p=0.11 according to the Wilcoxon test in the new version. Similar doubts arise for other indicators: SBP (p=0.13 and p=0.02, respectively), DBP (p=0.002 and p=0.001, respectively), BMI (p=0.52 and p=0.31, respectively), and PWV ( p=0.0001 and p=0.0001, respectively).

References:

https://en.wikipedia.org/wiki/Nonparametric_statistics

Author Response

I have reviewed the new submission of this manuscript and see that the authors are trying to refine the statistical section of their manuscript. However, I still see significant shortcomings in this section. Perhaps the authors need to involve a professional statistician to help refine the statistical processing of the results.

We thank the Reviewer for this additional opportunity to improve our manuscript.

Namely:

1. The authors noted that "Both Kolmogorov–Smirnov test and Shapiro–Wilk test revelaed that the measured variables were no normally distributed" and at the same time noted that "Data are presented as mean ± SD." It must be remembered that the "average arithmetic is used to represent scale variables with a normal distribution" and that "Standard deviation is intended to describe samples with a normal distribution and is not adapted to non-normal distributions." In a non-parametric distribution, other formats for representing quantitative data must be used: "The median, along with quartiles, is used to represent discrete variables or non-normally distributed quantitative continuous variables".

We have proposed a new Table 1 where median and interquartile renage values are presented. The corresponding lines in the Results section have been changed accordingly.

Please, see below for comparison between men and women.

2. The following statement of the authors also raises questions: "differences in mean values ​​between men and women were compared by non-parametric Wilcoxon test". First, with a nonparametric distribution, one does not have to talk about mean values ​​(see Remark 1). Secondly, the Wilcoxon test can also be used to check the differences between two samples of paired measurements (Wilcoxon signed-rank test), and to assess the differences between two independent samples in terms of the level of any trait measured quantitatively (The Wilcoxon rank-rum test or Mann-Whitney U test). Therefore, it is necessary to indicate which of the Wilcoxon tests the authors used.

We have specified in the Statistical analyses paragraph of the Methods we ahve adopted the Wilcoxon rank sum test or Mann Whitney U test. The corresponding p values are now presented in Table 1.

3. You should also consider the following feature of non-parametric statistical methods: "The wider applicability and increased robustness of non-parametric tests comes at a cost: in cases where a parametric test would be appropriate, non-parametric tests have less power. In other words, a larger sample size can be required to draw conclusions with the same degree of confidence".

In other words, when evaluating differences between groups, p-values ​​for non-parametric calculations should be markedly larger than when using Student's t-test. If we compare the results of the authors' calculations in the first version of the article and in the submitted manuscript, then such differences are not always noted. For example, the differences for the LF / HF ratio between men and women for were p=0.40 according to the Student's test and p=0.11 according to the Wilcoxon test in the new version. Similar doubts arise for other indicators: SBP (p=0.13 and p=0.02, respectively), DBP (p=0.002 and p=0.001, respectively), BMI (p=0.52 and p=0.31, respectively), and PWV ( p=0.0001 and p=0.0001, respectively).

We thank the Reviewer for the insightful comment. We have reported p values as indicated in the output of the SAS software per the specified PROC NPAR1WAY.

Christopher Morrell, Professor of Statistics at Loyola College in Baltimore (USA) and Senior Statistician at National Institue on Aging, NIH in Baltimore (USA) has guided and supervised us in these procedures.

Round 2

Reviewer 2 Report

This version contains almost no inaccuracies.

There is one small remark left:

Table 1 indicates that the results are presented as median values and interquartile range, while in the text of section 2.4. Statistical analysis was noted that "data are presented as mean ± SD". You need to eliminate this contradiction.

Author Response

Our apologies for having be "sloppy"

This manuscript is a resubmission of an earlier submission. The following is a list of the peer review reports and author responses from that submission.

Round 1

Reviewer 1 Report

In the Abstract and at the beginning of the Discussion the authors assert that “Beyond age, blood pressure, and diabetes, impaired cardiac autonomic function was a significant determinant of arterial aging”. Actually, as rightly pointed out by the authors among the limitations of the study, “the cross-sectional design does not allow determination of a causal relationship between PWV and cardiac autonomic”. This study simply showed an association between a vascular stiffness condition and impaired cardiac autonomic function. The Abstract and the Discussion need to be changed.

I recommend deleting the sentence “HRV may affect arterial stiffness by increasing the smooth muscle cell tone and, thus, decreasing arterial elasticity”, which is a hypothesis not supported by the results of the study and without a solid pathophysiological basis.

I would like to see the characteristics of the study population in relation to the study results, not only as a division between males and females (Table 1), also highlighting the role of cardiovascular treatment.

In Methods, paragraph 2.2. Assessment of Pulse Wave Velocity (PWV): “PWV was measured using the validated SphygmoCor device (AtCor Medical), whose validation and reproducibility have been previously published [23]”. However, the reference [23] does not refer to the validation of the SphygmoCor.

“PWV was calculated as the ratio of distance to transit time”. However, the algorithm used by SphygmoCor to calculate PWV does not use the direct distance between the registration site of the carotid and femoral pressure curve. I advise the authors to be more accurate in the description of the methods.

Author Response

In the Abstract and at the beginning of the Discussion the authors assert that “Beyond age, blood pressure, and diabetes, impaired cardiac autonomic function was a significant determinant of arterial aging”. Actually, as rightly pointed out by the authors among the limitations of the study, “the cross-sectional design does not allow determination of a causal relationship between PWV and cardiac autonomic”. This study simply showed an association between a vascular stiffness condition and impaired cardiac autonomic function. The Abstract and the Discussion need to be changed.

We thank the Reviewer for her/his careful revision of the manuscript and wise suggestions that contributed to make the manuscript more accurate and “scientifically sound”.

Concerning the specific comment, we have toned down this sentence.

The Discussion has been largely changed

I recommend deleting the sentence “HRV may affect arterial stiffness by increasing the smooth muscle cell tone and, thus, decreasing arterial elasticity”, which is a hypothesis not supported by the results of the study and without a solid pathophysiological basis.

We agree with the Reviewer. The sentence is no longer present in the revised manuscript.

I would like to see the characteristics of the study population in relation to the study results, not only as a division between males and females (Table 1), also highlighting the role of cardiovascular treatment.

We tried to better describe the specific study population in the Methods and in Table 1. We acknowledge it presents some limitations.

In Methods, paragraph 2.2. Assessment of Pulse Wave Velocity (PWV): “PWV was measured using the validated SphygmoCor device (AtCor Medical), whose validation and reproducibility have been previously published [23]”. However, the reference [23] does not refer to the validation of the SphygmoCor.

“PWV was calculated as the ratio of distance to transit time”. However, the algorithm used by SphygmoCor to calculate PWV does not use the direct distance between the registration site of the carotid and femoral pressure curve. I advise the authors to be more accurate in the description of the methods.

We apologize for the oversight. The Reviewer is totally right: we revised the description of PWV measurement by Sphygmocor

Reviewer 2 Report

I would like to thank you for the opportunity to read and review the article HEART RATE VARIABILITY DIFFERENTIALLY IMPACTS ON ARTERIAL STIFFNESS IN MEN AND WOMEN, which analyzes gender differences in the association of HRV and arterial stiffness in the presence and absence of DM.

When reviewing the article, I had certain comments.

1.     A reference to diabetes mellitus, the presence of which is an important factor studied in the article, should be added to the title of the article.

2.     In section 2.1. (and Table 1) a description of the diagnoses of the examined patients should be added. Table 1 indicates that 32% of patients had arterial hypertension, and 30.8% had diabetes. Even if we assume that these two nosologies had different patients, it remains unclear what the remaining 40% of patients suffered from?

3.     Section 2.4. it is necessary to add an indication - whether the data was checked for normal distribution and by what method. If the distribution was different from normal, then the use of Student's t-test was incorrect.

4.     Also in the same section, a description of the indicators included in the multiple linear regression model should be added.

5.     In table 1, a clear inaccuracy was made: in general, AH was detected in 32.1% of cases in the group, in women - in 51.9%, in men - in 70%. Which value is incorrect?

6.     Unfortunately, the authors did not take into account in their article many factors that can affect both HRV and arterial stiffness (smoking, the presence of dyslipidemia, the level of physical activity, the duration of diabetes [TODAY Study Group, 2022]], the therapy received for existing diseases, the presence of other diseases). I think that these indicators should be added to the manuscript.

7.     In table 2, the data presentation format is not clear; this needs to be specified more precisely.

8.     I do not understand the procedure for regression analysis. The authors write: «Based on the sex-differences reported in Table 1, multiple regression models were adjusted for age, hypertension, BMI, SBP, DBP, HR, RMSDD, LF, HF, LF/HF». Of these measures, age, SBP, and LF are presented in Table 2. Apparently, the other variables were also in the original model? If so, why are they not presented in the regression analysis table? For example, differences in the frequency of hypertension and the level of DBP in the groups of men and women are more pronounced (according to Table 1).

9.     The discussion section of the authors needs to be substantially improved. First, add links to recent articles on related topics (for example, TODAY Study Group, 2022, and 11. Rannelli LA, et al, 2017). Secondly, some of the references do not correspond to the statements of the authors of the manuscript in the discussion section. Thus, the study by Koenig J et al. studied healthy individuals, not patients with DM. The studies of Liatis S et al. and Chorepsima S (not Thorepsima S as in the manuscript) did not investigate predictive power for CV mortality and morbidity. And such discrepancies can be continued even further.

Referenses

TODAY Study Group, Shah AS, El Ghormli L, Gidding SS, Hughan KS, Levitt Katz LE, Koren D, Tryggestad JB, Bacha F, Braffett BH, Arslanian S, Urbina EM. Longitudinal changes in vascular stiffness and heart rate variability among young adults with youth-onset type 2 diabetes: results from the follow-up observational treatment options for type 2 diabetes in adolescents and youth (TODAY) study. Acta Diabetol. 2022 Feb;59(2):197-205. doi: 10.1007/s00592-021-01796-6.

Rannelli LA, MacRae JM, Mann MC, Ramesh S, Hemmelgarn BR, Rabi D, Sola DY, Ahmed SB. Sex differences in associations between insulin resistance, heart rate variability, and arterial stiffness in healthy women and men: a physiology study. Can J Physiol Pharmacol. 2017 Apr;95(4):349-355. doi: 10.1139/cjpp-2016-0122.

Chorepsima S, Eleftheriadou I, Tentolouris A, Moyssakis I, Protogerou A, Kokkinos A, Sfikakis PP, Tentolouris N. Pulse wave velocity and cardiac autonomic function in type 2 diabetes mellitus. BMC Endocr Disord. 2017 May 19;17(1):27. doi: 10.1186/s12902-017-0178-2.

Author Response

I would like to thank you for the opportunity to read and review the article HEART RATE VARIABILITY DIFFERENTIALLY IMPACTS ON ARTERIAL STIFFNESS IN MEN AND WOMEN, which analyzes gender differences in the association of HRV and arterial stiffness in the presence and absence of DM.

We thank the Reviewer for her/his careful revision of the manuscript and wise suggestions that contributed to make the message clearer, more accurate and readable.

When reviewing the article, I had certain comments.

1. A reference to diabetes mellitus, the presence of which is an important factor studied in the article, should be added to the title of the article.

We have modified the Title of the revised manuscript as suggested by the Reviewer .

2. In section 2.1. (and Table 1) a description of the diagnoses of the examined patients should be added. Table 1 indicates that 32% of patients had arterial hypertension, and 30.8% had diabetes. Even if we assume that these two nosologies had different patients, it remains unclear what the remaining 40% of patients suffered from?

We tried to better describe the specific study population in the Methods and in Table 1. We acknowledge it presents some limitations.

3. Section 2.4. it is necessary to add an indication - whether the data was checked for normal distribution and by what method. If the distribution was different from normal, then the use of Student's t-test was incorrect.

4. Also in the same section, a description of the indicators included in the multiple linear regression model should be added.

We have re-written the entrie paragraph about Statistical analyses trying to better describe the adopted models and the specific goal to achieve with that approach.

5. In table 1, a clear inaccuracy was made: in general, AH was detected in 32.1% of cases in the group, in women - in 51.9%, in men - in 70%. Which value is incorrect?

The Reviewer is right: it was a typo error (62.x rather than 32.x).

6. Unfortunately, the authors did not take into account in their article many factors that can affect both HRV and arterial stiffness (smoking, the presence of dyslipidemia, the level of physical activity, the duration of diabetes [TODAY Study Group, 2022]], the therapy received for existing diseases, the presence of other diseases). I think that these indicators should be added to the manuscript.

The Reviewer is right. We tried to better and briefly describe the complexity of factors affecting the relationship between HRV and PWV, also acknowledging the limitation of the present study for lack of information about some of these factors.

7. In table 2, the data presentation format is not clear; this needs to be specified more precisely.

8. I do not understand the procedure for regression analysis. The authors write: «Based on the sex-differences reported in Table 1, multiple regression models were adjusted for age, hypertension, BMI, SBP, DBP, HR, RMSDD, LF, HF, LF/HF». Of these measures, age, SBP, and LF are presented in Table 2. Apparently, the other variables were also in the original model? If so, why are they not presented in the regression analysis table? For example, differences in the frequency of hypertension and the level of DBP in the groups of men and women are more pronounced (according to Table 1).

We tried to better describe the statistical approach and the regression models. Accordingly, together with the paragrpah about Statistical analyses in the Methods Section, we modified the Results section and Table 2.

9. The discussion section of the authors needs to be substantially improved. First, add links to recent articles on related topics (for example, TODAY Study Group, 2022, and 11. Rannelli LA, et al, 2017). Secondly, some of the references do not correspond to the statements of the authors of the manuscript in the discussion section. Thus, the study by Koenig J et al. studied healthy individuals, not patients with DM. The studies of Liatis S et al. and Chorepsima S (not Thorepsima S as in the manuscript) did not investigate predictive power for CV mortality and morbidity. And such discrepancies can be continued even further.

Referenses TODAY Study Group, Shah AS, El Ghormli L, Gidding SS, Hughan KS, Levitt Katz LE, Koren D, Tryggestad JB, Bacha F, Braffett BH, Arslanian S, Urbina EM. Longitudinal changes in vascular stiffness and heart rate variability among young adults with youth-onset type 2 diabetes: results from the follow-up observational treatment options for type 2 diabetes in adolescents and youth (TODAY) study. Acta Diabetol. 2022 Feb;59(2):197-205.

Rannelli LA, MacRae JM, Mann MC, Ramesh S, Hemmelgarn BR, Rabi D, Sola DY, Ahmed SB. Sex differences in associations between insulin resistance, heart rate variability, and arterial stiffness in healthy women and men: a physiology study. Can J Physiol Pharmacol. 2017 Apr;95(4):349-355. doi: 10.1139/cjpp-2016-0122.

Chorepsima S, Eleftheriadou I, Tentolouris A, Moyssakis I, Protogerou A, Kokkinos A, Sfikakis PP, Tentolouris N. Pulse wave velocity and cardiac autonomic function in type 2 diabetes mellitus. BMC Endocr Disord. 2017 May 19;17(1):27. doi: 10.1186/s12902-017-0178-2.

We apologize for the naive mistakes in not properly revising that part of the Discussion with the related References.

The entire Discussion has largely been re-written and more relevant (as well as the proper) References have been quoted.

Round 2

Reviewer 2 Report

It should be recognized that the authors did a great job of eliminating comments on the text of the manuscript. However, they did not answer remark 3, which is fundamental:

3. Section 2.4. it is necessary to add an indication - whether the data was checked for normal distribution and by what method. If the distribution was different from normal, then the use of Student's t-test was incorrect.

Without a response to this remark, there is still no confidence in the scientific value of the results obtained by the authors.

Minor. What is source number 22 (SHARMANN)?

Author Response

We pologize for the oversihgt

The normality distribution is needed to use t-test in small samples. In larger sample, as it is the case of our population, is not needed.

We have tested for normality distribution and realized that the study variables were nto normally distributed.

As per Reviewer's suggestion we adopted teh non-parametric  Mann Whitney U test to compare mean values in men and women.

The manuscript has been modified accordingly

Reference 22: it should be correctly cited in the current version

We appreciate Reviewer's patience